# A multi-AUVs Bio-inspired Cooperative Hunting Algorithm for Environment with Ocean Current and Obstacles

Qingqin Liu
*Laboratory of Underwater Vehicles and Intelligent Systems*
*Shanghai Maritime University*
Shanghai, China
1125209092@qq.com

Bing Sun*
*Laboratory of Underwater Vehicles and Intelligent Systems*
*Shanghai Maritime University*
Shanghai, China
bingsun@shmtu.edu.cn

Chunhua Gu
*Research Institute Laboratory of Underwater Vehicles and Intelligent Systems*
*University of Shanghai for Science and Technology*
Shanghai, China
chgu@ecust.edu.cn

Dinghua Zhang
*CRRC SMD (SHANGHAI) Ltd*
Shanghai, China
zhangdh2@crrcsmd.com

Daqi Zhu
*Research Institute Laboratory of Underwater Vehicles and Intelligent Systems*
*University of Shanghai for Science and Technology*
Shanghai, China
zdq367@aliyun.com

*Abstract*—As a result of the complexity and indeterminacy of the underwater environment, multi-AUV cooperative hunting is a very challenging research area. In this paper, the focus is on the underwater environment with ocean currents and obstacles, and a novel multi-AUV bio-inspired cooperative hunting algorithm is presented. Firstly, a bio-inspired Neural Network model (BINN) is established to represent the AUV's underwater working environment. Each point in the grid map corresponds to the activity of a neural node in the BINN, and the next target point of the hunting AUV can be planned autonomously according to the neural activity of the neurons in the BINN. Secondly, the direction decision algorithm is embedded into the BINN hunting model for the ocean current environment, and the integrated algorithm proposed is compared with the hunting method without the direction decision. Finally, simulation studies are presented to demonstrate that the proposed algorithm is effective for cooperative hunting missions.

*Index Terms*—autonomous underwater vehicle; cooperative hunting; direction decision algorithm; bio-inspired neural network

## I. Introduction

Nowadays, multi-robot systems have been widely used in many areas [1]. For example, the multi-robot formation [2], cooperation hunting [3]–[5], task assignment and cooperation [6]–[8], etc. An autonomous underwater vehicle (AUV) is a robot that has intelligence and can complete tasks in the ocean by itself without the guidance of an operator. Because one

single AUV has limited capacity when working in the ocean, the multi-AUV system is an important research direction for the technology of current robots.

Many studies have been made on multi-AUV systems, including underwater cooperative search [9], [10], mine sweeping [11], autonomous navigation [12], formation control [13], dynamic task assignment [14], collaborative plotting [15], cooperative hunting [16]–[18] and so on. Among these studies, multi-AUV cooperative hunting is interesting, comprehensive, and worthy of study. But there are only a few researches on the underwater hunting problem now. Hunting in the ocean is different from the ground-hunting. Ding and Wang et al. proposed a virtual potential method for the path planning of AUV in [19]. Wang et al. [20] applied a potential field method to conduct the hunting of multi-AUV, where an ant colony optimization method is combined with an artificial potential field method to avoid collision in the hunting process. Zhu et al. [21], [22] studied the multi-AUVs hunting problem with the BNN algorithm, where the negotiation method is put forward to hunting task allocation. Simulation is completed in both 2-D and 3-D environments [23], and the proposed algorithm can deal with various situations automatically and catch the moving target [24] efficiently. However, these researches above has not considered the ocean current effects on the underwater environment. In fact, it is very important for multi-AUV cooperative hunting to consider the ocean current [25].

This paper focuses on situations in which the dynamic environment with ocean currents and obstacles as well as

This project is supported by the National Natural Science Foundation of China (62033009) and the Creative Activity Plan for Science and Technology Commission of Shanghai (23550730300).

the target is intelligent with their motions unpredicted and irregular. The multi-AUV hunting algorithm based on the bio-inspired neural network and the direction decision is presented. The hunting AUVs' paths are guided through a bio-inspired neural network and the direction decision model, and the results show that it can achieve the desired hunting result efficiency.

This paper is organized as follows: Section 2 describes the map of the hunting process, and the simulation of ocean currents and kinds of hunting final states are also given. In Section 3, the bio-inspired neural network algorithm and direction decision algorithm are designed. And the hunting algorithm is proposed in the environment of constant ocean currents. Simulations are conducted in Section 4, and Section 5 concludes the whole paper.

## II. PROBLEM STATEMENT

In this paper, the multi-AUVs hunting problem is studied in the underwater environment. Underwater environments replaced by discrete grid maps, the work area will be divided into the same small cells, each cell has two states: the free space and occupied space, as shown in Figure 1. The black cells indicate obstacles, and the white ones represent free space.

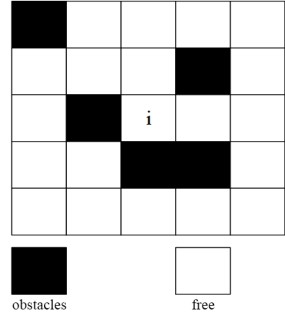

Fig. 1: Underwater grid map.

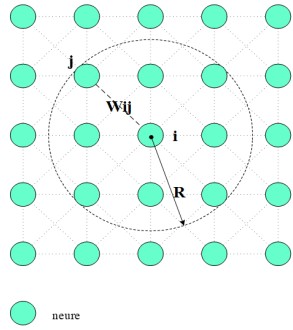

Fig. 2: 2D model of neural network.

The whole space of the hunting task is discrete, therefore, the hunting workspace can be used to simplify by grid map. The numbers of AUVs is denoted as $P_c = [P_{c1}, P_{c2}, \ldots]$, the evader is labeled as $Ev$, and the obstacles are denoted as $Obs = [Obs1 Obs2 \ldots \ldots]$. The evader has the same intelligent

abilities as the hunting AUVs, which can automatically avoid obstacles and update the location in real time to avoid being arrested.

A bio-inspired Neural Network model (BINN) is established to represent the AUV underwater working environment. Each neuron corresponds to the position in the grid map one by one in the neural network, as shown in Figure 2. Based on the distribution of neuron activity output in the neural network, plan the moving path of AUV. First, calculates all adjacent activity output of neurons to the current neuron, and then finds the maximum activity of these neurons as the next position of AUV. This model will help AUV to update the motion position in real-time and avoid obstacles automatically.

When the hunting task begins, the hunting AUV will move to the target, meanwhile, the target AUV updates the location as the escaped trajectory. In the process of hunting, hunting AUVs can avoid obstacles and find the shortest path between the AUV with the evader AUV. However, the evader needs to judge whether the contiguous positions are occupied by hunting AUVs or not. If not, continue to escape. If so, it needs to judge whether the surrounding positions are all occupied or not. If the positions are not occupied, determine a free space as the next step. Otherwise, stop moving and the hunting task is successful in the end. As shown in Fig.3 and Fig.4, four statuses of hunting the target successfully.

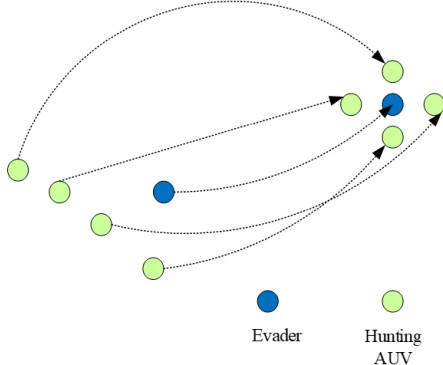

Fig. 3: The schematic diagram of the hunting process.

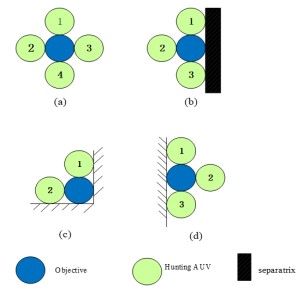

Fig. 4: Four kinds of hunting final states.

For the underwater environment, obstacles and ocean currents are very important factors that cannot be ignored. Obstacles are divided into static and dynamic obstacles, and the

ocean current is defined as the constant current with direction (45°) and speed (a grid each second).

## III. BIO-INSPIRED COOPERATIVE HUNTING ALGORITHM

Theoretically, AUV is a complex dynamic system that cannot move fully. However, the small turning radius does not affect the accuracy of the working space dimensions of AUV. Meanwhile, the size of AUV is much smaller than its working space in path planning, which can be thought of as a point. The ultimate principle of the path planning algorithm is on the basis of the grid map and establishes the biologically inspired neural network model, which represents the working environment. Each neural network neuron corresponds to the position in the grid map one by one. The connection between neurons to each other is shown in Figure 2. AUV will occupy a grid of two-dimensional environment at any moment, the adjacent 8 grids as the next optional positions of AUV. The specific navigation location depends on the activity of eight adjacent neurons and regards the maximum activity grid as the AUV navigation direction.

### A. Construction of the biological neural network

Dynamic model of neural distributions represented as a two-dimensional network of neurons, each neuron and its surrounding adjacent neurons are connected, as shown in Figure 2. Biologically inspired neural dynamics model can be converted to a shunting equation as follows [21]: The "shunting model" proposed by Grossberg is shown in the following formulation (1):

$$\frac{dx_i}{dt} = -Ax_i + (B - x_i)S_i^+ - (D + x_i)S_i^- \qquad (1)$$

where $x_i$ is the neural activity of the i-th neuron, $A$, $B$ and $D$ represent the passive decay rate, and the upper and lower bounds of the neural activity respectively which are nonnegative constants. $S_i^+$ and $S_i^-$ are the excitatory and inhibitory inputs to the neuron. In the process of hunting, the hunting AUVs' motions are guided by the dynamic landscape of a neural network. The excitatory input $S_i^+$ results from the target and its neighboring AUVs and the input $S_i^-$ only results from the obstacles. In this context, the dynamic of the i-th neuron in the neural network can be characterized by a shunting equation as

$$\frac{dx_i}{dt} = -Ax_i + (B - x_i)([I_i]^+ + \sum_{j=1}^{k} w_{ij}[x_j]^+) - (D + x_i)[I_i]^- \quad (2)$$

where $k$ is the number of neural connections of the i-th neuron to its neighboring neurons. The terms $[I_i]^+ + \sum_{j=1}^{k} w_{ij}[x_j]^+$ and $[I_i]^-$ are the $S_i^+$ and $S_i^-$ in equation (1), respectively. The terms $[I_i]^+$ is a linear-above-threshold function defined as $[I_i]^+ = max\{I_i, 0\}$, similarly the term $[I_i]^- = min\{-I_i, 0\}$. $[I_i]^+$ and $[I_i]^-$ are the variables that represent the input to the $i - th$ neuron from target and obstacle, respectively. They defined as:

$$I_i = I \begin{cases} E & \text{If it is a neighboring cell to objective} \\ 0 & \text{Otherwise(free space)} \\ -E & \text{If it is an obstacle} \end{cases} \quad (3)$$

where $E >> B$, which is a very large positive constant. In equation (2), the term $w_{ij}$ is defined as:

$$w_{ij} = f(|x_i - x_j|) \qquad (4)$$

where $x_i$ and $x_j$ are two vectors, is the Euclidean distance between them. Function $f(a)$ is a monotonically decreasing function which is defined as:

$$f(a) = \begin{cases} \mu/a & 0 < a < R_n \\ 0 & a \geq R_n \end{cases} \qquad (5)$$

where $\mu$ and $R_n$ are positive constants. Obviously, the weight connection coefficients are symmetrical, that is $w_{ij} = w_{ji}$. Fig. 2 shows the neural network in a 2-D environment model [12], where the green circles represent neurons. In this structure, each neuron is connected by adjacent ones which form the whole network for transmission of activity.

### B. The design of path decision

Discrete the navigation path of AUV, and the path planning problem will evolve into seeking the next position of AUV. Therefore, it is significant to link the specific dynamic environment with the preceding and next position of the AUV moments, and comprehensively the next driving position of the AUV. Using the distribution of the neuron activity output, find out the next time position of AUV. Firstly, calculate all the neurons adjacent to the current neuron activity output values, and then find out the maximum activity of these neurons, AUV began moving to the position where the largest output of the neural activity. When the same size of neuronal activity appears, the AUV will randomly select one of the neurons as their next position, which does not affect the process of path planning. The specific decisions of next position $x_{P_n}$ are:

$$\begin{aligned} \text{Path } &= \{P_n \mid x_{P_n} = \max\{x_i, i = 1, 2, \cdots, m\}, \\ &\quad P_p = P_c, P_c = P_n\} \end{aligned} \qquad (6)$$

$x_i$ represents the neuron's activity around the AUV, $P_p$ is the previous step of AUV, $P_c$ represents the current position of AUV, $P_n$ represents the next position. Under the process of AUV path planning, selecting the largest neural activity as its next movement repeatedly until reach the goal. As without considering the effect of ocean currents on AUV path planning, the introduction about ocean currents influencing the path planning of AUV will be expounded in the following section, and raise the corresponding solutions suitably.

### C. Hunting algorithm in the environment of ocean current

Under the effect of ocean currents, there is a certain deviation in the movement of AUV. In order to overcome the deviation and make AUV navigate as the predetermined trajectory, the direction decision mechanism is presented to solve this problem. During the process of path selection, reducing the ocean currents' effect on the navigational angle of AUV is the main ideal, it can be shown as formula (7):

$$\begin{aligned} \text{Path } &= \{P_n \mid x_{P_n} = \max\{x_i + g_i, i = 1, 2, \cdots, m\} \\ &\quad P_p = P_c, P_c = P_n\} \end{aligned} \qquad (7)$$

where $x_{P_n}$ means the active value of neurons surrounding at the current location, $P_p$ represents a previous step on the AUV, $P_c$ shows the current location of AUV, $P_n$ express the next location of AUV, $g_i$ is a function of an associated with a deflection angle which come from the ocean current effect, acting on AUV, which is defined as (8):

$$g_i = c * \cos \Delta \theta \tag{8}$$

$c$ is a constant greater than zero, setting the range as [0,1]. $\Delta \theta$ is the absolute value of the difference of $\theta_l$ and $\varphi_l$, defined as (9). $\theta_l$ denotes the angle that there is no current on AUV with a horizontal axis, defined as equation (10). $\varphi_l$ indicates the angle of AUV under the current direction and the horizontal axis, defined as (11).

$$\Delta \theta = |\theta_l - \varphi_l| \tag{9}$$

$$\theta_l = a \tan(y_{P_c} - y_{Pp}, x_{P_c} - x_{P_p}) \tag{10}$$

$$\varphi_l = a \tan(y_{Pn} - y_{Pc}, x_{P_n} - x_{P_c}) \tag{11}$$

$\theta_l$ indicates the angle of value currents direction and level axis, for currents size $a$, $(x_{P_p}, y_{P_p})$, $(x_{P_c}, y_{P_c})$ and $(x_{P_n}, y_{P_n})$ respectively express previous mobile location, current location and the final next mobile location, shown as Figure 5.

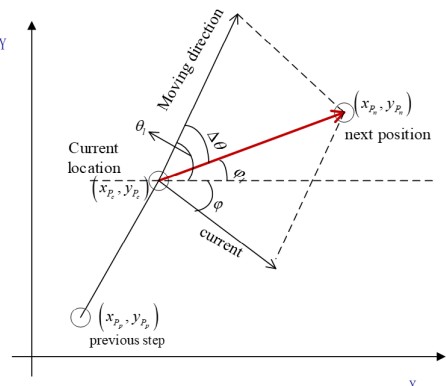

Fig. 5: The steering of AUVs in the ocean current environment.

When the currents direction is consistent with the scheduled mobile direction, the formula $\Delta \theta = 0$ and $\cos \Delta \theta = 1$ means AUV keeping movement direction with the scheduled direction. When the direction of the current is contrary to mobile direction with no currents effect instead, $\Delta \theta = \pi$ and $\cos \Delta \theta = -1$, after the path select of AUV in reduction off anti-direction of currents role and still keep the scheduled direction to path planning. Another situation uses direction decision-making to steer the direction of AUV under a constant current environment, shown as formula (7), which in order to accordance with the path planning of biologically inspired neural network, after losing steering effect of currents.

## D. The conflict resolution of hunting AUV

In view of the path conflict between AUV caused by neuron-selective conflict, as shown in Fig. 6, this paper proposes a solution called the location prediction method. This method records the location information between the hunting AUVs. Before each AUV selects the next moving position, if the moving grid is occupied by other AUV then choose the other grid to move. It can avoid the problem of path planning coordination between hunting AUVs effectively. The algorithm steps as follow.

Step 1: initialized the position of hunting AUVs.

Step 2: defines a vector type variable L to storage the initial position of each hunting AUV, and make a declaration in the program of AUV's path planning.

Step 3: updates the position information of AUV in variable L, the passed position has been clear.

Step 4: traverses the current position of other hunting AUV in variable $L$, and judges the position is equal with other position or not. If yes, then set this positional value as $-E$ and choose other grid as the next position of hunting AUV. If not, follow step 5.

Step 5: AUV finds the next position and store this position information in the variable $L$.

It can be seen that the next path selection will prejudge the location information of other AUV. If the grid to be moved has been occupied by other AUV, then the occupied grid can be seen as an obstacle and the hunting AUVs will not move towards this grid, avoiding the conflict between the AUVs. The simulation in Fig. 7 is that the hunting AUVs successfully resolute the conflict with the method proposed above.

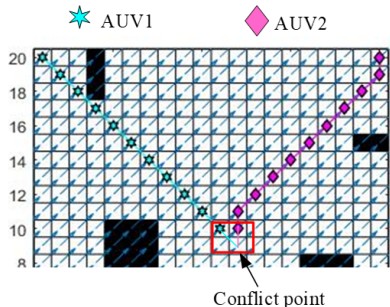

(a) The simulation of conflict

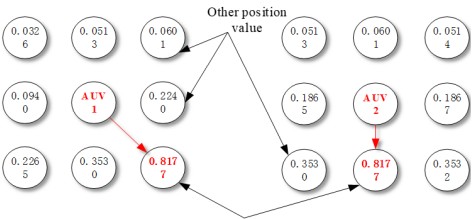

(b) The neuron activity value

Fig. 6: Conflict occurred.

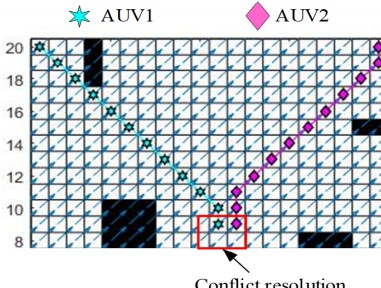

Fig. 7: The conflict resolution.

## IV. SIMULATION RESULT

In the two-dimension environment of hunting simulation, the hunting area can be defined as a set of $20 * 20$ grid map, the black grids are represented static or dynamic obstacles. In order to realize cooperative hunting mission, there are four hunting AUVs, which have the same intellectuality and can clearly communicate with each other. The evader with some intellectuality, whose maximum escape speed is 0.25 grid/seconds, while the speed of hunting AUV set 1 grid/seconds. The simulation parameters in the bio-inspired neural network model are setting as follow: $A = 2, B = D = 1, E = 100, mu = 0.7, c = 0.05, Rn = 2$.

### A. Cooperative Hunting with static obstacles

The angle of ocean current is $45°$, and the ocean current speed is about $\sqrt{2}$ gate grid/seconds. Red five-pointed star represents the evader and the remaining 4 different shapes are represented the hunting AUVs. The four AUVs in the hunting team, starting move respectively from (1,1), (20,16), (1,20) (17,7) four initial position to hunt the evader, whose initial position was (11,6).

Fig. 8 shows the different time state of the whole hunting task for static obstacles and ocean current environment. During the whole process of hunting, ocean currents will significantly affect the movement of AUVs, lengthen the hunting distance. We can see clearly that each hunting AUV was obviously influenced by the ocean current, moved more steps and cannot chose an appropriate path to complete the hunting task. The hunting trajectory of AUV was desultory which cannot pick out the concrete path of each hunting AUV accurately. During the whole hunting process, the evader AUV moved 5 steps and was hunt at (15,7) finally. Fig. 9 indicates the hunting simulation results with direction decision for static obstacles and ocean current environment. Fig. 9 shows the different time state of the whole hunting task. In the process of the hunting mission, each AUV selected the applicable path to hunt the evader AUV precisely. The evader moved 3 steps and was hunt at (11,9) finally. Compared with Fig. 8, the intention of each hunting AUV was more clearly and the paths of hunting AUV were obviously shortened after adding the direction decision mechanism, which can appropriately reduce the influence of ocean currents on AUVs. Under the same conditions, the direction decision mechanism can reduce

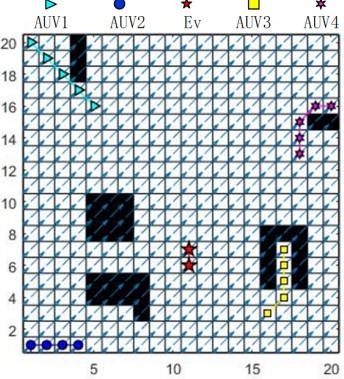

(a) the situation of hunting task at 4s

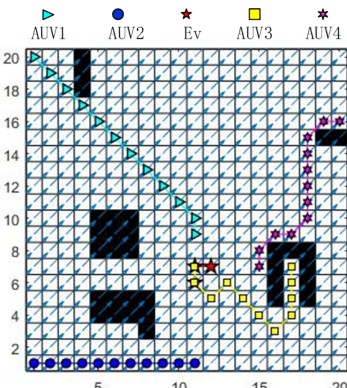

(b) the situation of hunting task at 10s

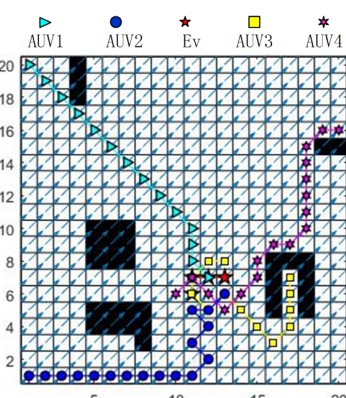

(c) the situation of hunting task at 16s

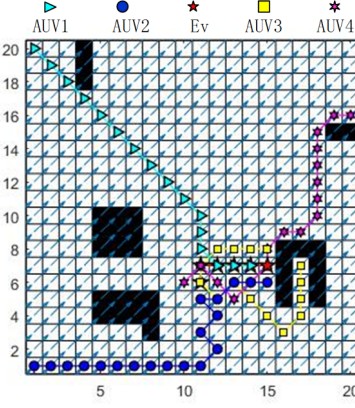

(d) the situation of hunting task at 18s

Fig. 8: Simulation for static obstacles and ocean current without direction decision scheme.

the steps of hunting AUVs that make the hunting task more efficient under the current setting. For instance, the path of AUV2 was clearly with pertinence and directive to move to the evader precisely. This direction decision mechanism can greatly reduce the effect from ocean current.

Simulation results of comparison experiments above are shown in Table 1. Sailing steps of each hunting AUV in the hunting process for different hunting algorithm are listed. As it is shown in Table 1, the proposed algorithm can help the AUVs to capture the evaders with less cost and higher efficiency. It takes the total steps of 40 for four hunting AUVs to catch the evader successfully. However, that of hunting algorithm without direction decision scheme is 73.

TABLE I: The steps of comparison in the ocean current environment with direction decision scheme or not

| AUV names | AUV | | | | Total steps | Evader |
|---|---|---|---|---|---|---|
| algorithms | 1 | 2 | 3 | 4 | | |
| with direction decision scheme | 10 | 10 | 11 | 9 | 40 | 3 |
| no direction decision scheme | 18 | 15 | 20 | 20 | 73 | 5 |

### B. Cooperative hunting with dynamic obstacles

The simulation above has shown the context of static obstacles with the decisions mechanism. The cooperative hunting with dynamic obstacles is simulated in here. Ocean current environment remains for 45°, size $\sqrt{2}$ grid/ seconds of constant current. Red five-pointed star represents the evader AUV and the other four different shapes are represented the hunting AUVs. The four AUVs in the hunting team, moving from (1,1), (20,16), (1,20) (17,7) respectively to hunt the evader. And the initial position of evader AUV was (11,6), the whole hunting process was shown in Figure 8 and Figure 9.

Fig. 10 shows the different time state of the total hunting task. During the process of hunting, ocean currents were distinctly affected the movement of AUVs, lengthen the hunting distance and prolong the time duration. Each hunting AUV was obviously influenced by the ocean current with more steps and cannot chose the shortest path to complete the hunting task. The hunting trajectory of AUV was desultory, it cannot find out the concrete path of each hunting AUV accurately. During the whole hunting process, the evader AUV moved 5 steps and was hunt at (14,7) finally, the trail of the moving obstacle started at (3,12) and it stopped at (9,12) ultimately.

Fig. 11 indicates the hunting simulation results with direction decision and shows the different time state of the whole hunting task. In the process of the hunting mission, each AUV selected the applicable path to hunt the evader AUV precisely. During the hunting process, as seen in Fig. 11, a moving obstacle passing through and just affect the path of AUV1, but cannot impact the final hunting task. The evader moved 5 steps and was hunted at (11,11) successfully, the trail of the moving obstacle stopped at (18,12) in the end.

In comparison with Fig. 10, the intention of each hunting AUV was more clearly and the paths of hunting AUV were obviously shortened after adding the direction decision mechanism, and can appropriately reduce the influence of ocean

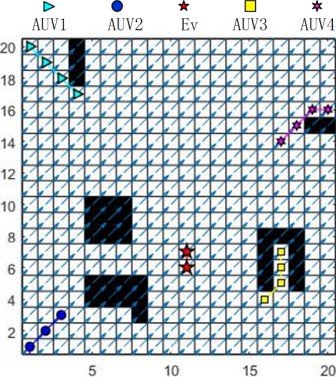

(a) the situation of hunting task at 1s

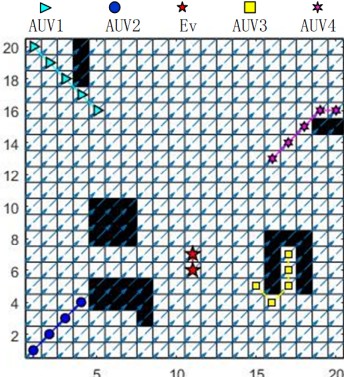

(b) the situation of hunting task at 4s

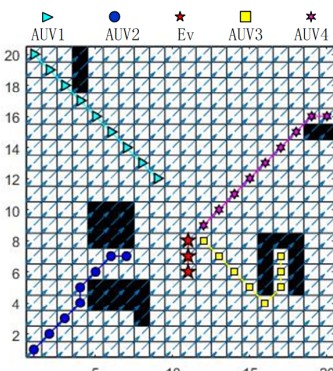

(c) the situation of hunting task at 8s

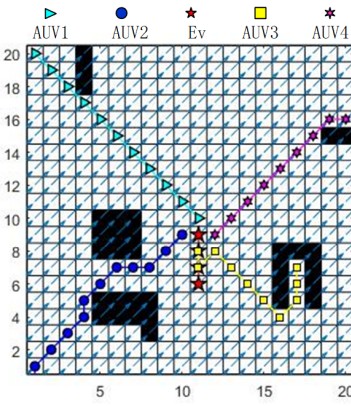

(d) the situation of hunting task at 12s

Fig. 9: Simulation for static obstacles and ocean current with direction decision scheme.

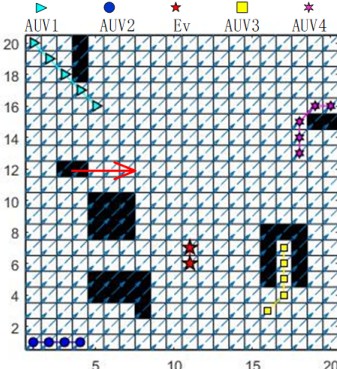

(a) The situation of hunting task at 4s

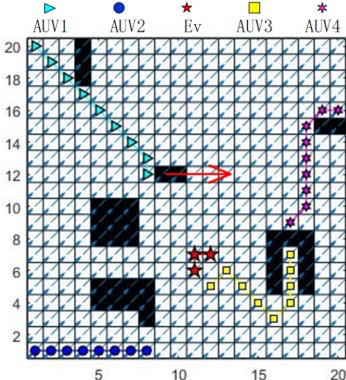

(b) The situation of hunting task at 8s

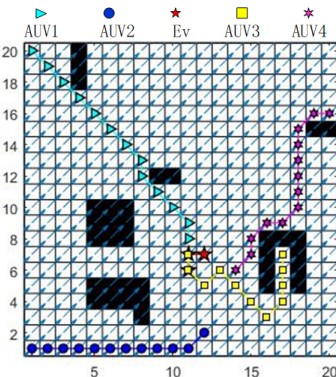

(c) The situation of hunting task at 12s

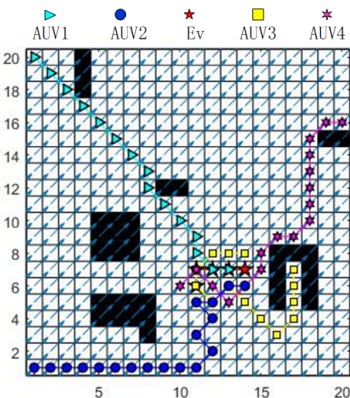

(d) The situation of hunting task at 18s

Fig. 10: The hunting task in dynamic obstacle environments without direction decision scheme.

currents on AUVs. Under the same conditions, the direction decision mechanism can reduce the steps of hunting AUVs that make the hunting task more efficient under the current setting. This direction decision mechanism can greatly reduce the effect from ocean current.

Results of comparison experiments about the different environment are shown in Table 2. The navigating steps of each AUV in the hunting task in both current and dynamic obstacle environments. As it is shown in Table 2, the algorithm raised above can greatly improve the efficiency of hunting AUVs. With the two algorithms, the steps of evader AUV were both 5 steps, the hunting total steps with direction decision scheme was 49 steps which was 10 steps lesser than the algorithm without direction decision scheme.

TABLE II: Steps of comparison in both current and dynamic obstacle environments

| AUV names | AUV | | | | Total steps | Evader |
|---|---|---|---|---|---|---|
| algorithms | 1 | 2 | 3 | 4 | | |
| with direction decision scheme | 14 | 17 | 13 | 15 | 59 | 5 |
| no direction decision scheme | 13 | 14 | 12 | 10 | 49 | 5 |

## V. CONCLUSION

The multi-AUV cooperative hunting problem is studied in this paper mainly. The direction decision mechanism based on bio-inspired neural network algorithm, can maintain the hunting path of each AUV unaffected by the ocean current and plan the effective path to hunt the evader AUV. The direction decision mechanism is proposed to reduce the influence from ocean current on AUV movement and improve the efficiency of hunting task. Because of the complicated underwater environment, the ocean current changes with the time actually which is a problem that need to be solved in the following research work.

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
