# OpenReview forum: "A multi-AUVs Bio-inspired Cooperative Hunting Algorithm for Environment with Ocean Current and Obstacles"
_IEEE.org/ICIST/2024/Conference — IEEE ICIST 2024 Conference Submission_

### Official Review · Reviewer_Vhro · 2024-08-22
**This article is very interesting and a good one**

**Rating:** 7
**Confidence:** 3

**Review:**

This study presented a novel multi-AUVs bio-inspired cooperative hunting algorithm in underwater environment with ocean current and obstacles. The obtained result is valuable and can be accepted if the following problems can be clarified.
 (1) In the introduction, it is not enough to state the current work. It should be expended and reconstructed.
(2) Where are ${[I_i^e]^ + }$, ${[I_i^o]^ + }$, ${[a]^ + }$ and ${[a]^ - }$  specifically reflected in formulas (1) and (2)? Please explain
(3) The simulation description can be simplified appropriately.
(4) There are some typos and grammar errors. The authors should have a native English speaker or software packages to perform the editing check.
(5) The conclusion of the article suggests using the present perfect tense for description.
(6) The references need to be updated and the format needs to be standardized.

---

### Official Review · Reviewer_HQsf · 2024-08-22
**he paper makes a valuable contribution to the field of autonomous underwater vehicles and multi-robot systems.**

**Rating:** 8
**Confidence:** 4

**Review:**

The work is significant as it addresses the complexities of multi-AUV cooperative hunting in challenging underwater environments, where factors like ocean currents and dynamic obstacles can greatly influence the success of the mission. The bio-inspired approach is innovative, and the inclusion of the direction decision algorithm adds robustness to the system. However, further validation in real-world scenarios would strengthen the findings, as the current study relies heavily on simulations. Additionally, exploring the algorithm's scalability to larger fleets of AUVs and more complex environments could provide deeper insights into its practical applicability. Overall, the paper makes a valuable contribution to the field of autonomous underwater vehicles and multi-robot systems. Below is a list of comments that should be taken into account further when revising the paper.

1.	There are a few typos in this paper which should be corrected. And there are some notions missed. Please make some corrections.
2.	Please add the necessary comments for Figures.
3.	Look out for the following grammatical and spelling errors: misspelling，redundant or unnecessary words，subject-predicate consistency problem，punctuation error，preposition error.

---

### Official Review · Reviewer_m8RU · 2024-08-23
**As the complexity and indeterminacy of the underwater environment, multi-AUV cooperative hunting is a very challenging research area. In this paper, it is focused on the underwater environment with ocean current and obstacles, and a novel multi-AUVs bio-inspired cooperative hunting algorithm is presented. Firstly, a bio-inspired Neural Network model (BINN) is established to represent the AUV underwater working environment. Each point in the grid map corresponds to the activity of a neural activity in the BINN, and the next target point of the hunting AUV can be planned autonomously according to the neuron activity of the neurons in the BINN. Secondly, the direction decision algorithm is embedded into the BINN hunting model for the ocean current environment, and the integrated algorithm proposed is compared with the hunting method without direction decision. Finally, simulation studies are presented to demonstrate that the proposed algorithm is effective for cooperative hunting mission. Comments for this submission are given as follows.**

**Rating:** 7
**Confidence:** 3

**Review:**

(1)、The pictures in this paper are clearly drawn, but there are some minor problems; the sizes of “obstacles” and “free” in Figure 1 are different, which should be revised by the authors, and the rest of the pictures should be double-checked by the authors.
(2)、The layout of this paper is very neat, although the authors are asked to revise the formatting of equation (7) if it is different from the formatting of the other equations at their discretion.
(3)、The literature citations in this paper are very appropriate, but whether the year of some of the literature is too old, for example, literature [15], the authors are requested to modify them at their discretion.

---

### Comment · Reviewer_Vhro · 2024-08-21
**This article is very interesting and a good one**

This study presented a novel multi-AUVs bio-inspired cooperative hunting algorithm in underwater environment with ocean current and obstacles. The obtained result is valuable and can be accepted if the following problems can be clarified.
(1) In the introduction, it is not enough to state the current work. It should be expended and reconstructed.
(2) Where are ${\left[ {I_i^e} \right]^ + },{\left[ {I_i^o} \right]^ + },{\left[ a \right]^ + }{\left[ a \right]^ - }$ specifically reflected in formulas (1) and (2)? Please explain$
(3) The simulation description can be simplified appropriately.
(4) There are some typos and grammar errors. The authors should have a native English speaker or software packages to perform the editing check.
(5) The conclusion of the article suggests using the present perfect tense for description.
(6) The references need to be updated and the format needs to be standardized.

---

### Decision · Program_Chairs · 2024-09-06

Accept (Oral)